

# Improving the quantification of peak concentrations for air quality sensors via data weighting

Caroline Frischmon*[1], Jonathan Silberstein*[1], Annamarie Guth[1], Erick Mattson[2], Jack Porter[3], and Michael Hannigan[1]

*Indicates both authors contributed equally and are both considered first author
[1]University of Colorado Boulder Department of Mechanical Engineering, 1111 Engineering Drive Boulder CO 80309
[2]Colorado Department of Public Health and Environment, 4300 Cherry Creek Drive South, Glendale, CO 80246
[3]South Coast Air Quality Monitoring District, 21865 Copley Drive Diamond Bar, CA 91765

**Correspondence:** Caroline Frischmon* (caroline.frischmon@colorado.edu)

**Abstract.** Traditional calibration models for low-cost air quality sensors have demonstrated a tendency to under-predict peak concentrations. We assessed the utility of adding data weights to low-cost sensor colocation data to improve the quantification of peak concentrations. Specifically, we explored the effects of data weighting on three different pollutant colocation datasets: total volatile organic compounds, carbon monoxide, and methane. Leveraging two different weighting functions, a sigmoidal

and piecewise weighting regime, we explored the impacts of the base model choice (multilinear regression vs random forest models), the sensitivity of weighting functions, and the ability of data weighting to improve high-concentration pollution measurements. When compared to unweighted colocation data, we demonstrate significant reductions in both error (root mean square error-RMSE) and bias (mean bias error-MBE) for pollutant peaks across all three datasets when data weighting is employed. For the top percentile of data, we observe an average of 23% reduction in RMSE and a 35% reduction in MBE

when optimal weights are employed. More significant reductions occurred in the 95-99th percentile of data, where MBE was reduced by an average of 70%. RMSE in the 95-99th percentile was reduced by an average of 26%. However, data weighting can also generate larger errors at baseline pollutant concentrations. Data weighting regimes were sensitive to input parameters, and input weighting functions may be tuned to better predict peak concentration data without significant reductions in the fidelity of baseline pollutant predictions.

## 1 Introduction

Over the past several decades, advances in both sensor technology and quantification methods have allowed for low cost sensor (LCS) networks to accurately quantify spatiotemporal changes in pollutant concentrations (Okorn and Hannigan, 2021a; Thorson et al., 2019; Karagulian et al., 2019). These sensors are often simple to deploy, making them well-suited to a wide variety of monitoring applications. For a fraction of the cost of a single regulatory monitor, several LCS may be deployed

across the monitoring area of interest-allowing for a more granular picture of local air quality (Morawska et al., 2018; Casey and Hannigan, 2018; Collier-Oxandale et al., 2018).



In many areas, sources of air pollution are often numerous, transient, and/or highly localized, requiring a wide suite of LCS to predict concentrations of relevant pollutants. Elevated concentrations of air pollutants such as volatile organic compounds (VOCs), methane ($CH_4$), and carbon monoxide (CO) can result in a series of adverse health and environmental outcomes (Zhou et al., 2023; Zhang et al., 2017). The health impacts of CO and VOC exposure include cardiopulmonary and respiratory illnesses and other forms of increased morbidity (Xing et al., 2016; Raub, 1999; Halios et al., 2022; Liu et al., 2022). $CH_4$ emissions result in significant contributions to the anthropogenic greenhouse gas budget, accelerating the impacts of climate change. Accurate assessment of the concentrations of these pollutants is essential in characterizing exposure, mitigating fugitive greenhouse gas emissions, and identifying areas of enhanced pollution.

LCS require extensive calibration to accurately predict air pollutant concentrations. As LCS signal may be affected by drift over time, changes in temperature and humidity, and cross-sensitivity to other pollutants, the raw sensor signal is often calibrated to ensure reliable measurements (Wei et al., 2018; Tancev, 2021; Rai et al., 2017; Sayahi et al., 2019; Mei et al., 2020; Jayaratne et al., 2018; Malings et al., 2020). Typically, LCS are calibrated via a procedure known as colocation-a process where sensors are placed adjacent to a reference grade monitor for an extended period. A regression model is developed to fit raw sensor signal to the measured concentrations of the reference monitor over the colocation period (Okorn and Hannigan, 2021b; Zamora et al., 2023). For large datasets spanning data from multiple sensors under varying air pollutant concentrations, researchers have developed correction algorithms to improve sensor signal during extreme events-though these correction factors require significant data and extensive testing (Barkjohn et al., 2021) Other methods for calibrating LCS, such as measuring sensor response to a target pollutant in a laboratory setting, may be poorly suited for field applications. The range of environmental parameters and chemical composition of the local environment may vary dramatically between a lab setting and field deployment of a LCS, resulting in large errors in predicted pollutant concentrations when lab calibrations are applied (Liang, 2021; Gonzalez et al., 2019).

Even in areas experiencing elevated air pollution, the majority of ambient LCS signal is often near-baseline, measuring relatively low pollutant concentrations (Collier-Oxandale et al., 2018; Casey et al., 2019; Sayahi et al., 2019; Bigi et al., 2018). LCS capture significantly elevated concentrations of air pollutants during short-term 'spikes' when sampling a plume from a pollution source (Mead et al., 2013). In the majority of LCS calibration models, both baseline data points and high concentration pollution spikes are given equal weights. However, as most ambient LCS data is near-baseline, calibration model solutions may generate coefficients that fit the baseline well, but result in large errors when predicting higher concentration pollution spikes (Okorn and Hannigan, 2021a). Furthermore, this unweighted calibration procedure may result in residuals that display significant biases at high concentrations, resulting in concentrations of predicted pollutants that are systematically lower or higher during concentration peaks than is physically accurate (Collier-Oxandale et al., 2018; Silberstein et al., 2024; Magi et al., 2020; Zimmerman et al., 2018).

For many LCS applications, including environmental justice analyses, industrial leak or event detection, wildfire plume quantification, and urban air pollution assessment, accurately quantifying pollutant spikes during elevated emission events is often the primary concern. To improve fitting of high concentration peaks during LCS calibration, additional model weights may be required. Data weighting is commonly employed in other environmental applications, including geospatial air pollution





analysis (Rose et al., 2009; De Mesnard, 2013) and hotspot identification (Bi et al., 2020). Other forms of data modification, such as data downsampling and upsampling, have been employed for LCS calibration. However, downsampling techniques often require considerable data, and the size of the final dataset employed in calibration is significantly downsampled (Furuta et al., 2022; Silberstein et al., 2024). Conversely, upsampling can often result in overfitting, as the technique adds duplicates for underrepresented data (Abhishek and Abdelaziz, 2023; Susan and Kumar, 2021).

In characterizing the performance of both calibration models and sensors during colocation, researchers employ statistical metrics such as the error and bias to assess the efficacy of LCS equipment (Casey et al., 2019; Sadighi et al., 2018). However, these metrics are calculated using the entire colocation dataset which may neglect for significant variability in error and bias across differing pollutant concentrations. As near-baseline pollutant concentrations comprise the majority of colocation data, large errors and biases during pollutant spikes typically have little effect on overall model performance. Consequently, researchers may overlook poor model fits at higher pollutant concentrations. Fields such as data science routinely employ weights to optimize for certain portions of a sample dataset. Our objective in this manuscript is to develop a methodology for the implementation of data weights in LCS applications, and to characterize the strengths and limitations of various weighting techniques. To our knowledge, this study represents the first application of LCS model weighting during colocation to improve sensor prediction of peak pollutant concentrations.

## 2 Methods

To study the efficacy of data weighting in sensor calibration models, we applied weights to three distinct datasets collected during low-cost sensor (LCS) colocations with reference instrumentation and compared the weighted model outcomes to the unweighted counterparts. For each colocation dataset, we tested two weighting schemes with two calibration model types to explore how various factors impact the weighted model predictions. The LCS data, reference data, weighting schemes, and calibration models are detailed below.

### 2.1 Instrumentation

LCS measurements were collected via HAQPods, a low-cost air quality monitor developed by the Hannigan Lab that integrates several commercially available sensors (Hannigan Lab, Boulder Colorado). Briefly, CO measurements were collected by an Alphasense CO-B4 sensor (Alphasense, Braintree United Kingdom), $CO_2$ measurements were collected by an ELT-S300 3.3V sensor (ELT Sensor Corp. Gyeonggido, Korea), and methane measurements were collected via an Figaro 2611 (Figaro, Rolling Meadows, Illinois) and MQ4 (Hanwei Electronics, Zhengzhou, China). The approximate cost, sensing range, and target pollutant of each sensor is given in Table S1.

### 2.2 Colocation Description

A single HAQPod was deployed for colocation at each of the three field sites across California and Colorado (Fig. 1). CO sensors were calibrated via an approximately ten-week field colocation with a reference 48i-TLE CO (Thermo Scientific)



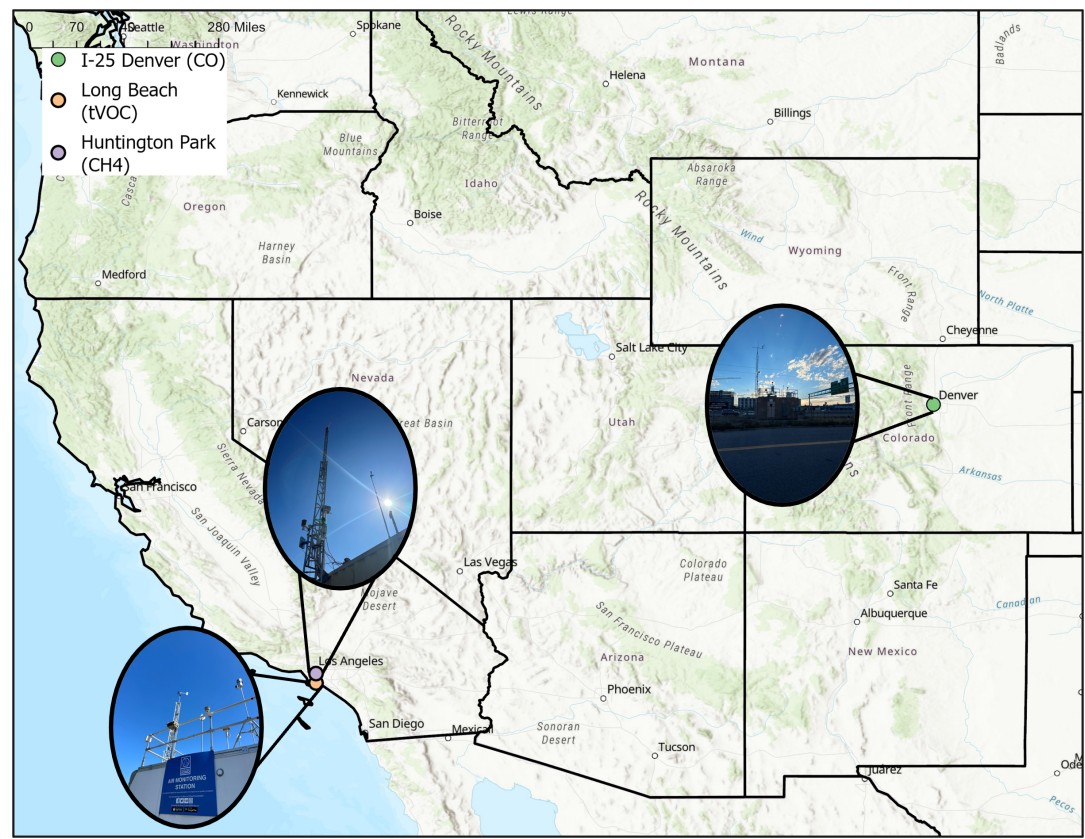

**Figure 1.** Colocation site map for the three study sites shown as colored circles. Images of each field site are displayed alongside each site.

monitor. The CO colocation field site was located directly adjacent to Interstate-25, a major highway (Fig. 1). Reference monitors were maintained by the Colorado Department of Public Health and Environment (CDPHE). $CH_4$ sensors were calibrated

via a 5-month colocation with reference Picarro $CH_4$-$H_2S$ Analyzer at a South Coast Air Quality Management District (South Coast AQMD) site in Huntington Park, CA (Fig. 1). TVOC sensors were calibrated with a reference FTIR continuous optical multi-pollutant analyzer (FluxSense Inc., San Diego, CA) for 5 months at a site in Long Beach, CA maintained by South Coast AQMD.

The LCS signal data was z-scored and time-averaged using median values in five minute increments. We also applied this

time-averaging to the reference data. Maximum reference CO concentrations over the colocation period were 1.84 ppm, and minimum reference concentrations were 0.04 ppm (Fig. 2). Maximum and minimum reference $CH_4$ concentrations were 8.75 ppm and 1.96 ppm. For TVOC, reference concentrations ranged from below the detection limit of 15 ppb to 2978 ppb.



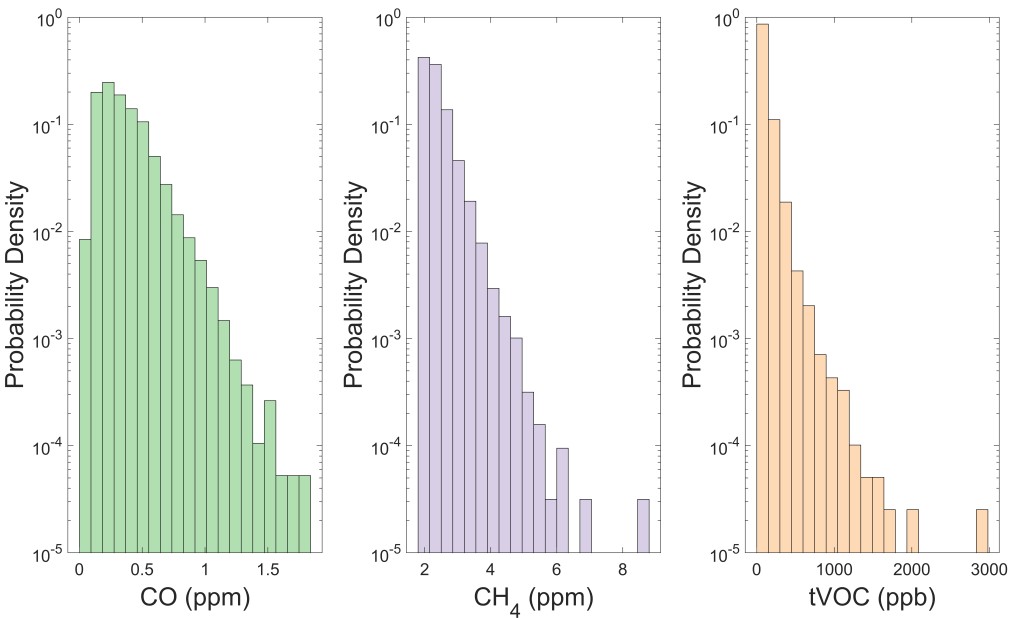

**Figure 2.** Reference measurement pollutant concentration distributions for CO (green), $CH_4$ (purple) and TVOC (orange) measurements. Probability density of each concentration bin is displayed on a log scale.

## 2.3 Calibration Models

### 2.3.1 Multilinear Regression (MLR)

MLR employs several independent variables to predict a dependent variable. In the context of LCS calibration, the independent variables correspond to individual z-scored sensor signals and the dependent variable corresponds to measured concentrations of an air pollutant by the reference instrument. Multilinear regression can be expressed as:

$$y_i = \beta_0 + \Sigma_{j=1}^p \beta_j x_{ij} + \epsilon_i \tag{1}$$

Where $\beta$s represent the model regression coefficients, y is the response for the $i^{th}$ observation, $x_{ij}$ is the $j^{th}$ predictor variable

for the $i^{th}$ observation, and $\epsilon_i$ represents the error term for the $i^{th}$ observation. For a weighted multilinear regression, the error term is defined as the weighted sum of squared residuals (SSR):

$$\text{Weighted SSR} = \Sigma_{j=1}^p W_i (y_i - \hat{y_i})^2 \tag{2}$$

Where $y_i$ is again the response for the $i^{th}$ observation, $\hat{y_i}$ is the predicted response for the $i^{th}$ observation, and $W_i$ is the weight applied to the $i^{th}$ observation in a weighted model. The model coefficients are tuned by minimizing the SSR.



MLR models are often employed in LCS applications when sensor behavior is linear over a specific range of air pollutants and environmental parameters. MLR models are well-suited for small training datasets, as they tend to overfit training data less than machine learning models.

### 2.3.2 Random Forest (RF)

RF models represent a more complex alternative to MLR models. These models have previously been employed in higher
complexity LCS datasets that contain extensive colocation data (Zimmerman et al., 2018). These models are composed of several decision trees, and the final RF model prediction is influenced by the predictions of each of the individual trees. As an ensemble model, RF predictions are less likely to be overfit than a single prediction from any individual decision tree. Machine learning models such as RFs require tuning of hyperparameters including the number of decision trees, number of estimators, and the minimum leaf size. Hyperparameters were determined for each pollutant colocation dataset via the minimization of the
root mean squared error. Additional information regarding RF models can be found in other manuscripts (Zimmerman et al., 2018; Karagulian et al., 2019). The error term (weighted RMSE) for a weighted random forest model is defined in Eq. 3.

$$\text{Weighted RMSE} = \sqrt{\frac{\sum_{i=1}^{n} W_i (y_i - \hat{y}_i)^2}{\sum_{i=1}^{n} W_i}} \tag{3}$$

### 2.4 Data Weighting Functions

To assess the strengths and limitations of different methods for weighting colocation data, we tested two variants of weighting
functions on our data. Assessed weighting functions are displayed in Fig. 3. These functions are discussed in detail below.

### 2.4.1 Piecewise Weights

The simplest weighting method explored in this study leverages piecewise weights as shown in Eq. 1. Reference data ($y_i$) above a certain percentile, X, were given a weight of 1, while data below the percentile were given a weight of 0.1. In Sect. 2.6, we tested the sensitivity of our piecewise weighting scheme to percentile set points ranging from the 75-99.9th percentile.

$$W(z) = \begin{cases} .1 & z \leq X \\ 1 & z > X \end{cases} \tag{4}$$

### 2.4.2 Sigmoidal Weights

Secondly, a sigmoidal squared offset weighting distribution given in Eq. 2 was applied. Reference data were then assigned weights as:

$$W(z) = \frac{1}{(1 + e^{-z+X})^2} \tag{5}$$




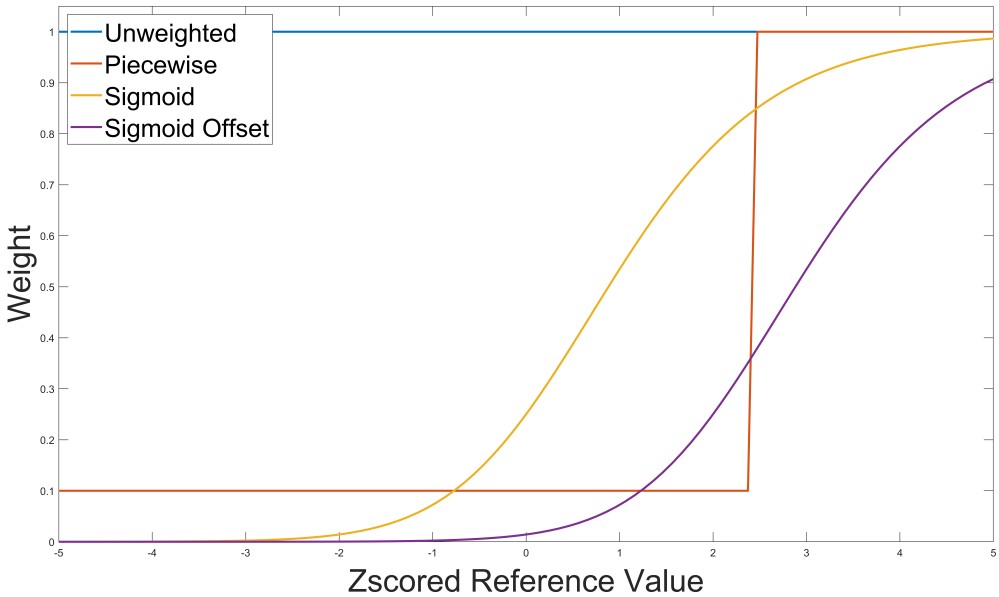

**Figure 3.** Weighting functions employed in this study as a function of z-scored reference measurement. Example unweighted (blue), piecewise (orange), sigmoidal (yellow), and sigmoidal with an offset (purple) weighting curves are shown above.

Where z represents the z-scored value of a reference monitor concentration, and x is the user input offset for the function. The offset, X, shifts the weighting distribution to higher or lower z-scored values depending on user preference. In Sect. 2.6, we analyzed the model sensitivity to offsets ranging from 0-3.

## 2.5   Data Analysis Frameworks

The LCS predicted concentrations were segregated into percentile ranges, based on the corresponding reference concentrations,
to better understand the impacts of data weighting on low, average, and high (peak) concentration data. Data were grouped between the 0-5th percentile, 5th-25th percentile, 25th-75th percentile,75-95th percentile, 95th-99th percentile, and 99th-100th percentiles. Model performance was assessed as a function of mean bias error and root mean squared error. As RF models displayed lower biases and errors when compared to MLR models, these machine learning models will be discussed in detail in subsequent sections. Fitting statistics for MLR models are displayed in tables S2-S4.

## 2.6   Sensitivity Analyses and Colocation Application

To understand how our regression solutions change as the parameters of the weighting functions are altered, we conducted a sensitivity analysis. We varied the linear offset or percentile cut off, x, of the sigmoidal and piecewise weighting function, respectively, to understand how predicted air pollutant concentrations changed. To conduct this sensitivity analysis, we varied





the offset in intervals of 0.5 between values of 0 and 3.0 for sigmoidal weights, and varied the percentile cut-off to the 75th,
80th, 85th, 90th, 95th, 99th, 99.5th, and 99.9th percentiles for piecewise weights.

Assessing the uncertainty associated with predicted air pollutant concentrations is often complex, as errors and biases may
be influenced by a combination of sensor performance, model choice, and the distribution of pollutant concentrations. For each
of the previously discussed colocation datasets, we plotted the RMSE and MBE as a function of reference data concentration
percentile to understand the sensitivity of errors and biases to changes in weighting offsets. In subsequent analyses, we further
investigated the weighting functions that improved high-concentration fitting.

Using the optimal weighting functions, we then investigated how our predicted concentrations changed over the field colo-
cation when data weighting was implemented.

## 3 Results and Discussion

### 3.0.1 CO Sensitivity Analysis and Colocation Application

As shown in Fig. 4 below, predicted values of CO in piecewise testing data were similar for reference concentrations less
than 1 ppm, and displayed significantly reduced error and biases at elevated concentrations when compared to the base MLR
and RF models. Piecewise MLR weighting functions outperformed baseline MLR and weighted and unweighted RF models
at elevated concentrations of CO, with minimal degradation in performance at lower concentrations (Fig. 4, Table S2-S4).
Sigmoidal weighting functions displayed increased sensitivity to input parameters at lower concentrations of CO and displayed
lower RMSE and greater biases than piecewise weights. These results indicate assessing multiple input parameters to model
weighting functions is essential in determining the optimal weights and weighting scheme for a colocation data set. When
compared to unweighted models, optimal weighting schemes were able to better quantify high concentrations of CO. We
further analyzed the sigmoidal $z_{sigmoid}$=2, and percentile$_{piecewise}$ = 95th as these functions displayed minimal enhancements
in error at baseline values while better predicting high concentration data.

As data weighting schemes are implemented (Fig 5), our model is able to more accurately quantify peaks-both in testing and
training data. Sigmoidal weighting systematically biases CO measurements, elevating the baseline above actual concentrations
(Fig. 5). As a result of this biasing, the linear line of best fit between predicted and reference concentrations is shifted when
compared to the unweighted MLR model, resulting in poorer fits at low concentrations of CO. Piecewise weights were able
to better predict high concentration CO episodes without systematically skewing lower concentration measurements (Fig. 5).
These techniques allow for improved quantification of high-concentration emissions events from urban sources of CO, such as
motor vehicle usage and industrial facilities.

### 3.0.2 CH$_4$ Sensitivity Analysis and Colocation Application

Weighting sensitivity analysis for CH$_4$ indicated that RF models are better suited for data weighting with the CH$_4$ data, as
baseline data in the MLR models showed significantly increased RMSE and MBE with weighting (Fig. 6). RF models showed



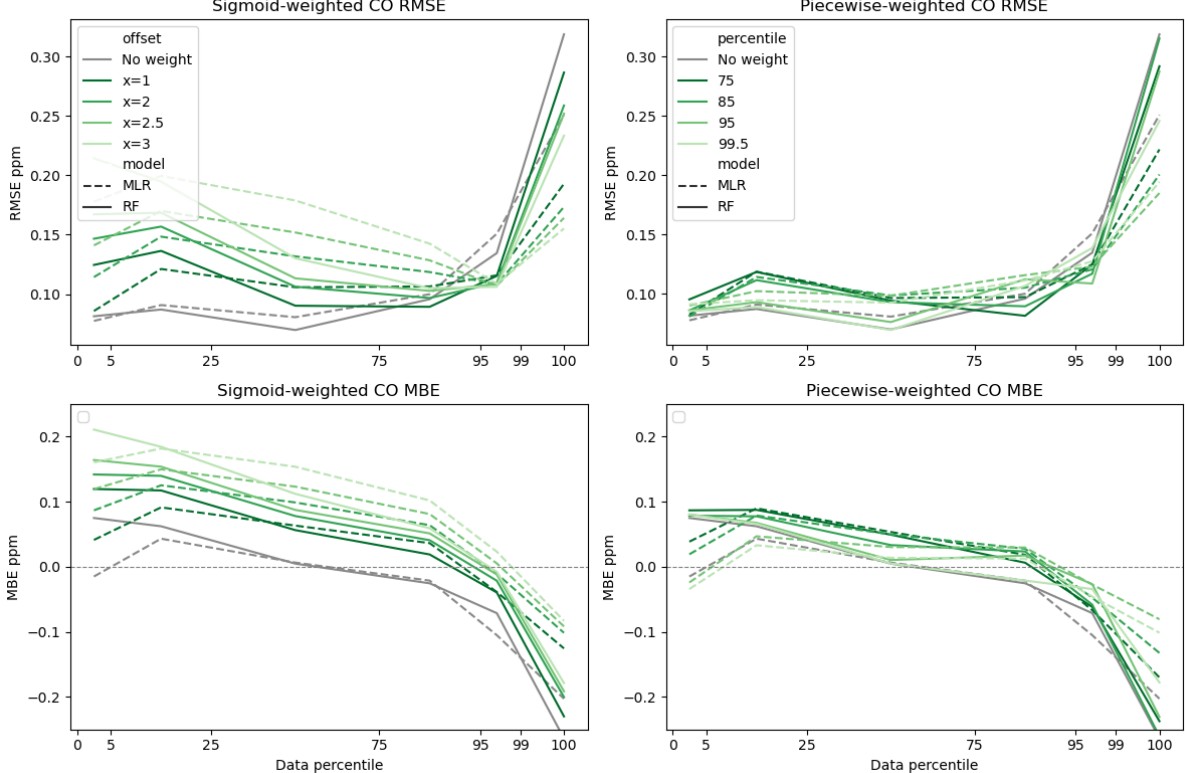

**Figure 4.** CO sensitivity to weighting parameters. RMSE and MBE are displayed as a function of data percentile for unweighted data as well as for piecewise and sigmoidal weights. Dashed lines indicate MLR fits whereas filled lines represent RF fits. Lighter colors indicate increased offsets for weighting distributions.

less impact on data fits at the baseline, while still decreasing RMSE and MBE in data above the 95th percentile. As with CO, sigmoid-weighting generally performed better for data above the 95th percentile but caused worse fits at the baseline than piecewise weighting. Reductions in RMSE and MBE generally increased as the offset (sigmoidal-weighting) and percentile cut-off (piecewise-weighting) values increased until the 95th percentile. Therefore, we chose to further analyze the sigmoidal $z_{sigmoid}$=3 and percentile$_{piecewise}$ = 95th.

RMSE reductions in data above the 95th percentile ranged from 16-28% for the chosen weighting models. Sigmoidal $z_{sigmoid}$=3 weighting reduced the MBE of data in the 95-99th percentile by 72%, compared to 47% using piecewise weighting. In both models, reductions in MBE were not as large for data above the 99th percentile (32% for sigmoidal and 20% for piecewise) compared to data in the 95th-99th percentile. Fig. 7 illustrates how weighted models underpredict the highest concentrations of $CH_4$ (above approximately 5 ppm) even as the models better predict elevated concentrations below the highest

percentile.



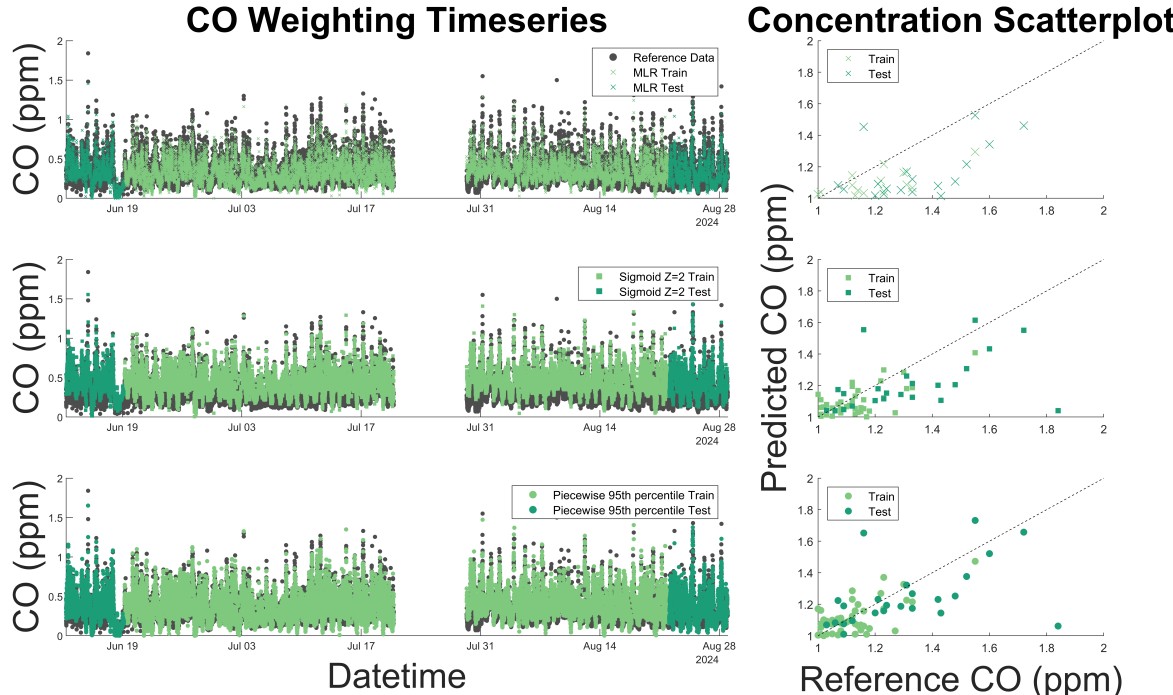

**Figure 5.** CO timeseries and 1:1 scatter plot for unweighted (x) and optimal sigmoidal (square) and piecewise (circle) functions superimposed on reference data. The 1:1 line is displayed as a dashed black line on the scatter plot. Dark colors represent testing data and light colors represent training data.

### 3.0.3 TVOC Sensitivity Analysis

In the TVOC weighting sensitivity test, piecewise-weighting models were less sensitive to weighting changes across MLR and RF models compared to sigmoid-weighting models (Fig. 8). In subsequent analysis, we focus on the $\text{percentile}_{piecewise}$ = 95th model because it best reduced MBE above the 95th percentile of data. We chose to further investigate sigmoidal $z_{sigmoid}$=3 model because it reduces MBE nearly to zero in the 95-99th percentiles. However, sigmoidal $z_{sigmoid}$=3 also caused elevated bias at lower concentrations. In applications where lower concentration fits are relevant, sigmoidal $z_{sigmoid}$=1 may be a better choice.

Approximately 29% of the TVOC reference data is at or below the method detection limit, so RMSE and MBE in the 0-5th and 5-25th percentiles are not shown in the figures. Compared to $CH_4$ and CO, the TVOC data showed the greatest difference in high concentration fits between sigmoidal and piecewise weighting. For example, the MBE for data in the 95-99th percentile and 99-100th percentile was reduced by 97% and 32%, respectively, using sigmoidal weighting, compared to just 40% and 16% using piecewise weighting. Sigmoidal weighting simultaneously penalizes baseline data while prioritizing high concentration data, whereas piecewise weighting only weights high concentration data while treating baseline and interquartile data the





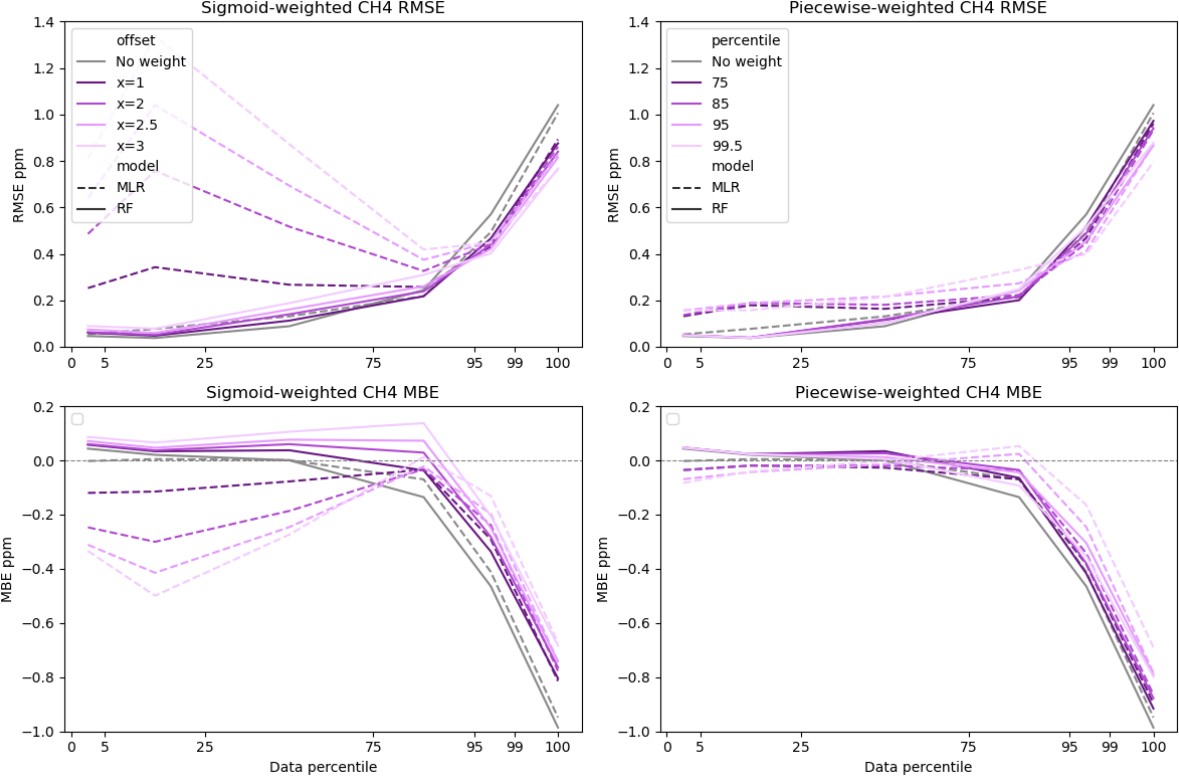

**Figure 6.** $CH_4$ sensitivity to weighting parameters. RMSE and MBE are displayed as a function of data percentile for unweighted data as well as for piecewise and sigmoidal weights. Dashed lines indicate MLR fits whereas filled lines represent RF fits. Lighter colors indicate increased offsets for weighting distributions.

same. In a dataset where 29% of the data is below the detection limit, it appears there is an added benefit to downweighting the
baseline data using sigmoidal weighting.

## 3.1 Discussion General Trends

Leveraging data weighting in colocation datasets, we found reductions in both bias and error across peak CO, $CH_4$ and TVOC concentration measurements, especially in the 95-99th percentile of the reference concentrations (Fig. 10). While the 99-100th percentile data still showed reduced bias and error across all three datasets, the reductions were less significant in the TVOC and
$CH_4$ data, as indicated by the magnitude of the arrows in Fig. 10. This diminished improvement in $99^{th+}$ percentile error and bias may be due to the enhanced variation between data in the top percentile of $CH_4$ and TVOC datasets, which respectively varied between 3.7-17.6 and 3.6-30.9 standard deviations above the mean (Fig. 2). The top percentile of CO data varied between 3.3-8.3 standard deviations above the mean, which accounts for significantly less high-concentration variability than for the





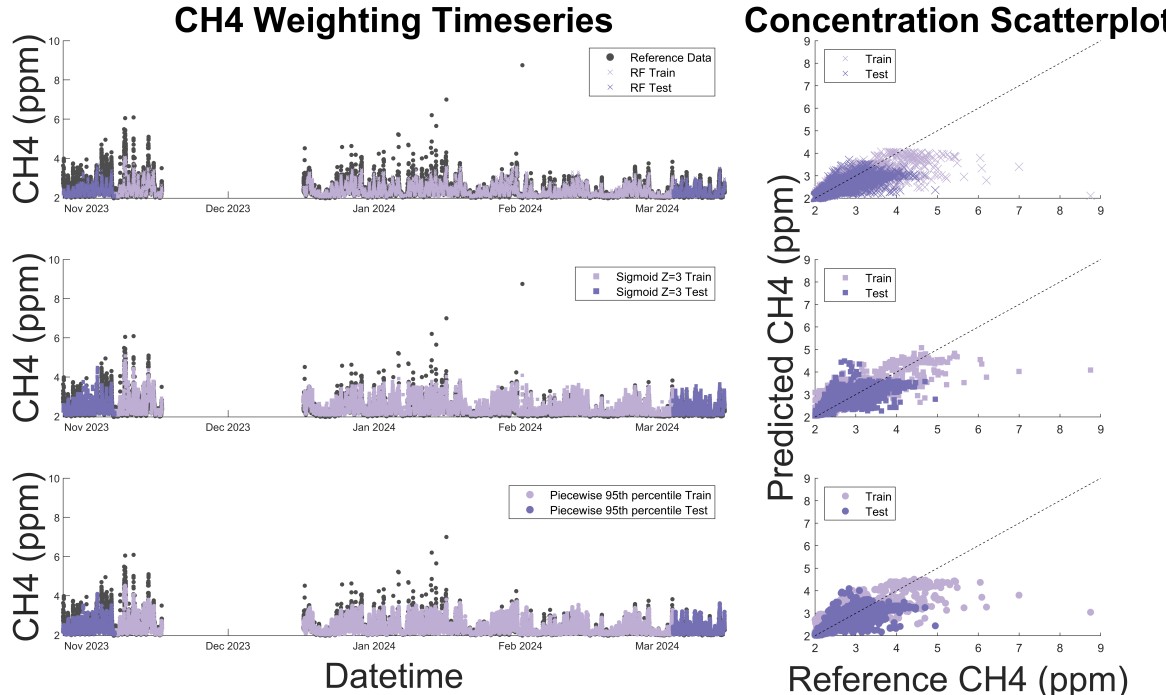

**Figure 7.** CH$_4$ timeseries and 1:1 scatter plot for unweighted (x) and optimal sigmoidal (square) and piecewise (circle) functions superimposed on reference data. The 1:1 line is displayed as a dashed black line on the scatter plot. Dark colors represent testing data and light colors represent training data.

other datasets (Fig. 2). Increased variability in pollutant concentrations within the top percentile may result in poorer fits as

even with data weighting regression models may struggle to fit extreme outliers.

Though we observed an improvement in our fitting of high-concentration samples, select data points still displayed large discrepancies between reference and predicted concentrations, especially in the highest percentile of data. These results highlight that data weighting methodologies may not ameliorate all causes of concentration under-prediction, such as large systematic error as a result of sensor degradation or the LCS not observing the plume. In these cases, further improvements in sensor

development and colocation techniques are required. We also note that the large error and bias at the highest percentile of data is not evident in the overall RMSE and MBE values across all three models (indicated with a black 'X' in Fig. 10). This is likely because the RMSE and MBE in lower concentration data masks the high bias and error of the relatively sparse elevated concentrations. Data partitioning during error analysis proved essential here for fully understanding the performance of the colocation models.

While the sigmoidal weighting scheme was more effective at improving colocation fits for data above the 95th percentile, it also introduced more bias and error at the lower percentiles than the piecewise weighting scheme (Fig. 10). The sigmoidal function assigns a continuous distribution of weights that both penalizes baseline data and prioritizes high-concentration data




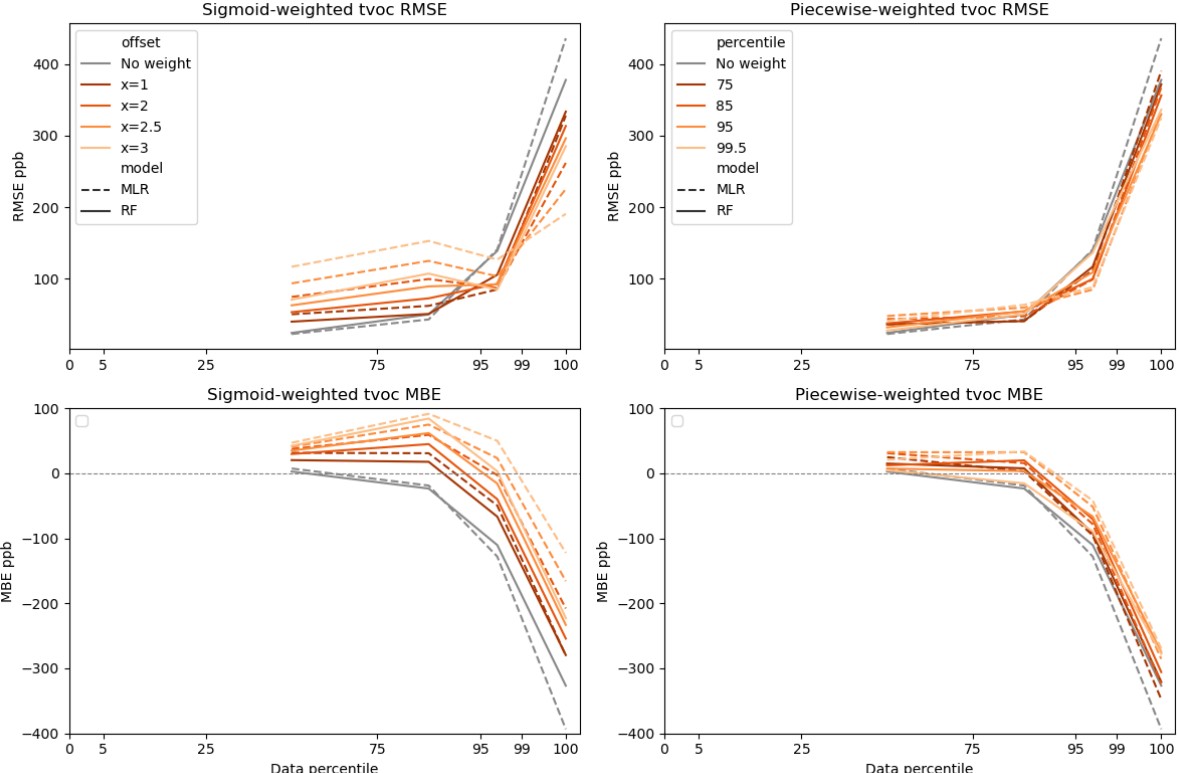

**Figure 8.** TVOC sensitivity to weighting parameters. RMSE and MBE are displayed as a function of data percentile for unweighted data as well as for piecewise and sigmoidal weights. Dashed lines indicate MLR fits whereas filled lines represent RF fits. Lighter colors indicate increased offsets for weighting distributions.

(Fig. 3). Conversely, the piecewise function prioritizes high-concentration data while treating baseline and interquartile data the same. The trade-off observed in the sigmoidally weighted data may be worthwhile in certain applications where accurate prediction of elevated concentrations is pertinent but error at the baseline is acceptable, such as in pollution event detection and plume prediction (Clements et al., 2024; Kanabkaew et al., 2019; **?**).

The optimal regression model and weighting parameters were unique for each colocation dataset. Even as the datasets showed similar patterns in the sensitivity analysis, each individual pollutant datasets displayed differential sensitivities to changes in input weighting parameters. For example, the unweighted random forest model showed a better fit than unweighted multiple linear regression for CO. However when applying data weights, the weighted random forest model overfit the CO data, leading to greater reductions in bias and error for the weighted multiple linear regression models. In the TVOC and $CH_4$ datasets, random forest performed better in both the weighted and unweighted models. These results underscore the importance of testing different weighting schemes on a colocation dataset, as a single 'one-size-fits-all' approach may unnecessarily elevate error at low concentrations and marginally improve fitting at higher pollutant values.





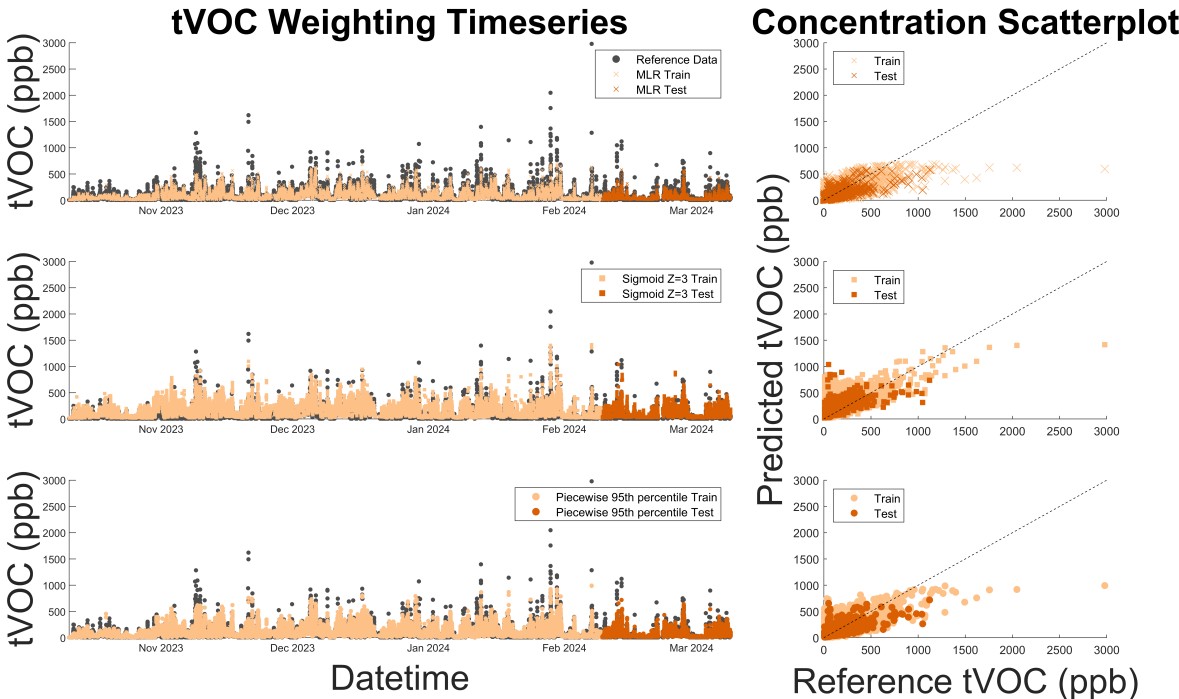

**Figure 9.** TVOC timeseries and 1:1 scatter plot for unweighted (x) and optimal sigmoidal (square) and piecewise (circle) functions superimposed on reference data. The 1:1 line is displayed as a dashed black line on the scatter plot. Dark colors represent testing data and light colors represent training data.

## 4 Conclusions

Employing model weights in ambient gas-phase colocation datasets, we found that data-weighting improved our ability to quantify elevated air pollutant concentrations. We assessed the performance of unweighted, sigmoidally weighted, and piecewise weighting schemes in MLR and RF models for CO, TVOC, and $CH_4$ calibration. MBE was reduced by 16-97% and RMSE was reduced by 13-39% in concentration data above the 95th percentile when model weights were applied. Our error analysis also underscored the importance of examining statistical metrics for colocation models within partitioned percentile groups, as the high error and bias of peak concentration (> 95th percentile) predictions were not evident in the error and bias statistics of the overall models.

Model predictions for weighted CO, TVOC, and $CH_4$ colocation models were sensitive to changes in our weighting functions-indicating a systematic assessment of optimal weighting parameters is required in each colocation dataset to best identify those that improve high-concentration fitting performance. The optimal regression model and weighting parameters varied between colocation datasets, which reflected the variability in pollutant distributions and sources. Generally, sigmoidal weighting improved high concentration predictions more than piecewise weighting, but sigmoidal also introduced larger errors and bias





**Figure 10.** CO, CH$_4$ and TVOC target plot displaying the RMSE and MBE for different data percentiles (marker size). Lighter colors represent RF models and darker colors represent MLR models. Optimal unweighted (x), piecewise (circle), and sigmoidal (square) weighting functions are displayed for each pollutant. The black X indicates the MBE and RMSE of the overall unweighted model and arrows indicate the direction of improvement at the 95-99$^{th}$ percentiles and 99$^{th+}$ for the weighted compared to unweighted optimal models.

for baseline data. The optimal model weight scheme will depend on the data application, as for some applications, such as emissions event detection, accurate prediction of elevated concentrations should be prioritized over baseline concentrations.

.



In the future, we plan to explore how baseline concentration accuracy might be preserved in a weighted model using a hybrid approach that combines the strengths of both unweighted and sigmoidal weighting schemes.

*Data availability.* Calibrated sensor fits and statistics are included in Tables S2-S3. Table S1 includes sensor descriptions and specifications, Tables S2 contains summary statistics for CO concentrations, Table S3 contains summary statistics for $CH_4$ concentrations, and Table S4 contains summary statistics for TVOC concentrations.

*Author contributions.* Conceptualization: Caroline Frischmon and Jonathan Silberstein; methodology: Annamarie Guth, Caroline Frischmon, Erick Mattson, Jack Porter, Jonathan Silberstein; software: Annamarie Guth, Caroline Frischmon, Jonathan Silberstein; formal analysis: Annamarie Guth, Caroline Frischmon, Jonathan Silberstein; investigation: Michael Hannigan; resources: Erick Mattson, Jack Porter, Michael Hannigan; writing and editing: Annamarie Guth, Caroline Frischmon, Jonathan Silberstein; visualization: Annamarie Guth, Caroline Frischmon, Jonathan Silberstein; supervision: Michael Hannigan; project administration: Michael Hannigan; funding aquisition: Michael
Hannigan.

*Competing interests.* The authors declare no competing interests. This research was funded by NIEHS 1R01ES033478 and ASPIRE NSF ERC under grant number 1941524.

*Acknowledgements.* Thank you to the Colorado Department of Public Health and Environment and the South Coast Air Quality Monitoring District for reference instrument data, as well as the Utah Department of Environmental Quality for testing initial colocation setups. Thank
you to the Hannigan Lab Dev Team (Percy, Spencer, Sasha, Pete) for help troubleshooting and assembling air quality monitors.





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
