# Peer review of "Improving the quantification of peak concentrations for air quality sensors via data weighting"

_EGUsphere, 2024_

## Author Response (AR1)

We are grateful for the feedback received from the reviewer and have responded to specific comments below. In order to improve the clarity of our figures, we added some context to the figure captions and moved some figure details to the supplementary information.

- What TVOC sensor was used?

  - VOC measurements were collected via Figaro metal oxide sensors now listed in lines 84-85 in the text.

- It is hard to look at figures 4 and 6 and understand which performs best

  - Thank you for pointing this out. We have updated the plots to only include the best model type (MLR or RF) for each pollutant to make these plots easier to read. Plots for both MLR and RF are now available in the supplemental information.

- It would be helpful to understand what the concentrations are associated with the data percentiles.

  - Thank you for the suggestion. This information was added to the captions of Figures 4, 6, and 8.

- Testing/training is not described in the methods

  - We appreciate the referee bringing this gap to our attention. We added the information shared below on testing/training to lines 144-146.

    - "For CO and CH4, the first and last ten percent of data was used excluded from model training to test the models' ability to predict concentrations under unseen conditions. This data is hence referred to as testing data. For the TVOC dataset, the last 20 percent was used as testing data to achieve more peaks within the testing dataset.}"

- Line 231 "?" in the citation

  - We have corrected this citation issue. Thank you.

We appreciate the reviewer's feedback on our study. In particular, we found their definition of the pollutants as those that "do not vary much diurnally and present as rare, intermittent transient events with the vast majority of data being at some baseline level" especially helpful and have added part of this definition to the abstract.

Below are our responses to the reviewer's specific comments.

1. In section 2.1 there is no mention of the TVOC sensors make/models used.

- o VOC measurements were collected via Figaro metal oxide sensors now listed in lines 84-85 in the text.

2. I noticed throughout most of the paper and figures, sigmoidal weighting is discussed/appears before piecewise weighting, so consider switching the order of Sections 2.4.1 and 2.4.2 to be consistent.

- o Thank you for this suggestion. We have switched the orders for consistency.

3. Both sections 2.4.1 and 2.4.2 use a "X" variable to describe either a percentile or an offset, which is confusing. Consider using a different variable other than "X" to describe one of those. In addition, Section 2.6 uses lowercase "x" instead of uppercase "X" by mistake.

- o We changed the percentile to P and fixed the lowercase "x" in Section 2.6. Thank you!

4. In section 3, the subsection numbering is unusual (e.g. 3.0.1 rather than 3.1 or 3.1.1); consider using nonzero subsection numbering.

- o We appreciate the referee bringing this to our attention and have changed the numbering to 3.1.

5. In general I find Figs. 4-9 not easy to decipher.

For the sensitivity plots, perhaps there are too many weighting parameters shown on the same plot, but I find it difficult to see which weighting parameters are performing best in

order to connect it with the in-text statements of which weighting parameters were further explored (e.g., "Therefore, we chose to further analyze the sigmoidal z_sigmoid=3 andn percentile_piecewise=95th").

For the timeseries/scatterplots, I likewise am having trouble distinguishing whether the sigmoidal or piecewise scatterplots are hugging the 1-1 line closer. I think there could be some refinement of Figs 4-9 to help make it clearer on how the reader can also arrive to the in-text conclusions.

- o Thank you for pointing this out. We have updated Figures 4, 6, and 8 to only include the best model type (MLR or RF) for each pollutant to make these plots easier to read. Plots for both MLR and RF are now available in the supplemental information. For Figures 5, 7, and 9, we added some information to the figure captions to help readers understand the conclusions that can be drawn from these plots. Regarding the comparison between sigmoidal and piecewise for the 1-to-1 line, we agree that it is not easy to distinguish the difference between each weighting scheme here. This is why we choose to compare sigmoidal and piecewise using Figure 10 rather than Figures 5, 7, and 9. The only exception to this is the apparent shifted baseline for CO using sigmoid weighting.